# Field Effectiveness of Drones to Identify Potential *Aedes aegypti* Breeding Sites in Household Environments from Tapachula, a Dengue-Endemic City in Southern Mexico

**DOI:** 10.3390/insects12080663

**Published:** 2021-07-21

**Authors:** Kenia Mayela Valdez-Delgado, David A. Moo-Llanes, Rogelio Danis-Lozano, Luis Alberto Cisneros-Vázquez, Adriana E. Flores-Suarez, Gustavo Ponce-García, Carlos E. Medina-De la Garza, Esteban E. Díaz-González, Ildefonso Fernández-Salas

**Affiliations:** 1Centro Regional de Investigación en Salud Pública (CRISP), Instituto Nacional de Salud Pública (INSP), 4a Av. Norte esquina 19 Calle Poniente s/n, Tapachula 30700, Chiapas, Mexico; kenia.valdez@insp.mx (K.M.V.-D.); david.moo@insp.mx (D.A.M.-L.); rdanis@insp.mx (R.D.-L.); luis.cisneros@insp.mx (L.A.C.-V.); 2Facultad de Ciencias Biológicas, Universidad Autónoma de Nuevo León (UANL), Ave. Pedro de Alba s/n cruz con Ave. Manuel L. Barragán, San Nicolás de los Garza 66455, Nuevo León, Mexico; adriana.floressr@uanl.edu.mx (A.E.F.-S.); gustavo.poncegc@uanl.edu.mx (G.P.-G.); 3Centro de Investigación y Desarrollo en Ciencias de la Salud, Universidad Autónoma de Nuevo León, Av. Mutualismo, Monterrey 64460, Nuevo León, Mexico; carlos.medina@uanl.mx; 4Laboratorio Estatal de Salud Pública, Servicios de Salud de Nuevo León, Serafín Peña 2211, Guadalupe 67180, Nuevo León, Mexico; esteban.diaz@saludnl.gob.mx

**Keywords:** *Aedes aegypti*, drone, breeding sites, mosquito surveillance, Tapachula

## Abstract

**Simple Summary:**

Mexico’s mosquito control program requires better tools for effective mosquito surveillance in dengue-endemic areas. Additionally, technology must be more available in order to gain a better understanding of the factors that condition the presence of mosquitoes in residential settings. This study is the first to report on the use of drones for mosquito breeding surveillance in Mexico and aims to evaluate the effectiveness of low-cost drone images in order to identify Aedes aegypti mosquito breeding sites on the roofs of buildings and backyards. The results were compared to the current government Aedes vector surveillance program, which is based on on-ground activities in Tapachula city, Chiapas, southern Mexico. Through drone surveillance, we identified a total of 983 containers that were distributed in 10 types; approximately one-third (2752) of the containers were inspected by ground surveillance systems that were distributed in 26 container types. The concordance between drone and ground surveillance was 64.9% in detecting at least one container using both methods. Drones can identify the most common roof containers and should be used in dengue-endemic areas that have many possible breeding sites and are in accessible housing areas. Drones can be useful in complementing mosquito surveillance and control programs.

**Abstract:**

*Aedes aegypti* control programs require more sensitive tools in order to survey domestic and peridomestic larval habitats for dengue and other arbovirus prevention areas. As a consequence of the COVID-19 pandemic, field technicians have faced a new occupational hazard during their work activities in dengue surveillance and control. Safer strategies to monitor larval populations, in addition to minimum householder contact, are undoubtedly urgently needed. Drones can be part of the solution in urban and rural areas that are dengue-endemic. Throughout this study, the proportion of larvae breeding sites found in the roofs and backyards of houses were assessed using drone images. Concurrently, the traditional ground field technician’s surveillance was utilized to sample the same house groups. The results were analyzed in order to compare the effectiveness of both field surveillance approaches. Aerial images of 216 houses from El Vergel village in Tapachula, Chiapas, Mexico, at a height of 30 m, were obtained using a drone. Each household was sampled indoors and outdoors by vector control personnel targeting all the containers that potentially served as *Aedes aegypti* breeding sites. The main results were that the drone could find 1 container per 2.8 found by ground surveillance; however, containers that were inaccessible by technicians in roofs and backyards, such as plastic buckets and tubs, disposable plastic containers and flowerpots were more often detected by drones than traditional ground surveillance. This new technological approach would undoubtedly improve the surveillance of *Aedes aegypti* in household environments, and better vector control activities would therefore be achieved in dengue-endemic countries.

## 1. Introduction

Co-circulation of dengue (DENV), Zika (ZIKV) and Chikungunya (CHIKV) viruses in tropical and subtropical America is a major public health concern. More than 50 million dengue fever (DF) cases occur each year worldwide, resulting in ≈24,000 deaths annually [1]. In Mexico, ≈550,000 cases of DF and ≈165,000 cases of severe dengue were reported from 1990 to 2020 [2]. The mosquito *Aedes aegypti* is the primary vector of dengue virus and it has evolved to mate, feed, rest, and lay eggs in and around urban and rural households. This species shows diurnal rhythms of feeding, and the highest biting activity is observed in the mid-morning and late afternoon. Females frequently bite multiple human hosts during each gonotrophic cycle [3].

Traditional entomological surveillance was carried out by the National Vector Control Program by the Ministry of Health in Mexico. One component that has been used in the program is larval indices, which have been proposed by the World Health Organization since the 1960s, and include the house (or premise) index, the container index and the Breteau index [4,5]. These indicators have shown little correlation with adult densities in premises and their values are frequently not associated with the number of dengue cases in an endemic locality [6,7,8,9,10,11]. To increase the sensitivity of vector surveillance, the National Vector Program has incorporated the pupal index since 2008. The objective of this component is to identify the most productive water container types in order to design more targeted and cost-effective vector-control interventions [12]. Additionally, other entomological indicators, such as the percentage of positive ovitraps, eggs per ovitrap, eggs per block, and percentage of positive blocks to *Aedes aegypti* and/or *Aedes albopictus,* were also included in the program [13].

There is clearly a need to integrate modern technology in order to quickly and accurately monitor housing conditions, which would allow for an increase in the effectiveness of mosquito surveillance programs. Since the beginning of 2020, the simultaneous occurrence of *Aedes*-transmitted diseases and COVID-19 in Latin American countries has presented a life-threatening risk for field technicians involved in *Aedes aegypti* surveillance and control activities. This new risk has demanded the incorporation of changes and innovation strategies in order to reduce personnel being exposed to COVID-19 during household visits [14,15]. Furthermore, the drone industry has developed rapidly in recent years, and has extended drone use to commercial applications that are separate from military and recreational domains [16,17]. The use of unmanned aerial vehicles (UAVs) offers the ability to collect detailed real-time spatial information at a relatively low cost and avoids many of the limitations associated with satellite data (e.g., long repeat times, cloud pollution, low spatial resolution, lack of homogeneity in camera angle and shooting time) [18]. Some attempts have been made using this technology, e.g., obtaining images by aerial shooting and further analyzing them in order to identify and to map the presence of *Aedes aegypti* and *Culex quinquefasciatus*. These maps can support the surveillance and control measures and/or can be used in communication materials to display specific risk conditions for a given area [19].

This study was conducted in the city of Tapachula, which is located in southern Mexico, and aims to evaluate the effectiveness of low-cost drone images to identify *Aedes aegypti* mosquito breeding sites on roofs and backyards. Similarly, the results were compared to current government *Aedes* vector surveillance programs based on ground activities.

## 2. Materials and Methods

### 2.1. Study Area

Tapachula (320,451 inhabitants) is in the southern Pacific coast of Chiapas state at the southern border of Mexico–Guatemala. The weather patterns show an average annual temperature of 21.7 °C and rainfall of 1000–5000 mm, with a heavy rainy season prevailing from May to November. The chosen study area was the locality of El Vergel (14°56′25.13″ N, 92°15′53.66″ W), which is a neighborhood in the northwest of Tapachula, with an area of 433 km^2^ (Figure 1) and an estimated population of 4290 people (residents and workers) in 592 homes [20].

### 2.2. Breeding Habitats Sampling by Traditional Ground Surveillance (GS)

Field surveys, which included visits to households from El Vergel, were conducted during the rainy season, from 13–20 August 2019 (Figure 1). Each dweller was asked to sign an informed consent in order to approve their participation in the project entitled “Use of drones for the association of risk factors with the abundance of mosquitoes *Aedes aegypti* Linnaeus (Diptera: Culicidae), in dengue transmission areas of Tapachula, Chiapas”. This ground surveillance was specifically planned and conducted as part of this study.

First, the entomological surveillance was conducted by field technicians who searched for all water-holding containers, as well as containers capable of holding water and becoming mosquito-breeding sites (made of plastic, steel, glass, pewter, and clay). Different locations of the household, including indoor, backyard (outdoor), and roofs (when it was possible), were inspected by technicians. Then, the containers were sorted according to their type, utility, shadow type, location, water volume and if they were positive for the presence of *Aedes* immatures (eggs number, larvae per stage and pupae). The complete classification of containers was published previously [21].

### 2.3. Breeding Habitats Inspections by Drone Surveillance (DS)

A low-cost drone model Phantom 4 Pro DJI^®^ was utilized, which integrates a 20-megapixels camera with a focal length of 35 mm and a theoretical resolution of 1 cm. The flight logistics followed national regulations, referred to as CO AV 23/10 R3 regulation of Dirección General de Aeronáutica Civil de la Secretaría de Comunicaciones y Transportes of Mexican Government [22], and the user’s guide from the manufacturer’s recommendations [23]. We used the DJI GO 4^®^ app for manual mode flight [24]. Photographs of each house were taken at a height of 30 m and at around 12.00 p.m. in order to minimize the shadow effect, as this can hamper object identification [25,26,27]. Each drone battery was sufficient for a 12–15 min flight, which allowed us to take photos of 15 to 20 houses per day. These photos were taken in the same survey week and covered the roofs and backyards of surveyed households. The inhabitants consented to the photographs being shot over their dwellings through their signing of an informed consent form previously. Once the aerial photographs were obtained, the limits of each property were verified by colony cartography with ArcGIS 10.3 software. The images were saved in JPEG format and processed on the free software GNU Image Manipulation Programme (GIMP^®^ 2.8) [28,29,30]. The potential breeding sites were manually searched and counted using GIMP software [28]. A database of the characteristics of containers was built according to the characteristics of the survey. The containers were counted if they held or had the potential to hold water, which could result in the development of *Aedes aegypti* larvae.

### 2.4. Data Analysis

The effectiveness of ground surveillance sampling was defined as the technician’s ability to locate both household indoor and outdoor containers, with and without water that could serve as a breeding site. The effectiveness of drone imagery was defined as the ability to identify and record potential *Aedes aegypti* breeding habitats, and specifically covered the outdoor environment, such as roofs and backyards. The data were expressed as the number of containers detected in roofs, backyards, and the total. We compared and evaluated the concordance between the identification of the different types of containers by drone imagery and the traditional methodology of ground surveillance that expressed the percentage of households with the presence of at least one container detected by drone surveillance (DS) and ground surveillance (GS). Additionally, Kendall’s tau statistic was performed in order to identify whether certain kinds of containers were located more often via ground-based inspection or via drones using STATA 14 with a significance of 0.05 (StataCorp, College Station, TX, USA).

## 3. Results

### 3.1. Ground Surveillance (GS) Method

A total of 2752 containers (water-holding or containers capable of holding water), distributed in 26 container types, were found in 216 houses by the ground surveillance approach (Table 1). On average, a mean of µ = 12.7 containers per dwelling were calculated, showing that 77.3% of them were concentrated in seven types of containers: flowerpot (31.6%), non-disposable plastic containers (10.3%), plastic buckets and tubs (9.1%), disposable plastic containers (8.2%), cement washbasins (large) (7.4%), cans (5.5%), and animal drinking containers (mobile) (5.2%). The rest of the 19 container types represented only 22.7%. All of the 2752 containers were sorted based on their household environment location, which accounted for 2334 in the outdoors (84.8%) and 418 indoors (15.2%). Outdoors yielded a higher number of pooled containers, 2334 (84.8%), while only 418 (15.2%) were identified indoors (Table 1).

Only 177 (6.4%) of pooled indoor, plus outdoor wet containers, showed *Aedes aegypti* larvae. The frequency of larval populations was higher at household indoors (14.1%), as compared to containers found outdoors (5.1%). Overall, technicians reported that indoor containers with higher larvae findings were large cement washbasins (43.3%) and 55 Ga drums. These same containers also had more mosquito larvae when located at household outdoors, accounting for 34.3% (large washbasin) and 21.8% (55-Ga drums), respectively. A group of 38 elevated storage tanks were the only roof container type seen by ground surveillance because 78.7% (170) of dwellings had metal roof sheets, and thus surveying them from ceilings was a fruitless activity.

In situ operations by field technicians allowed for additional data recording of ground larval containers. For instance, 995 (42.6%) out of 2334 outdoor containers were localized in backyard open areas, 734 (31.5%) under vegetation or trees, and 605 (25.9%) under shadows, respectively. In addition, field personnel found 1167 (42.4%) containers that held water, 329 at indoors and 838 in the backyard (outdoor). Water volume for indoor containers averaged 480.8 L of water versus 131.9 L of backyard containers; most of these were frequently used for storing water within homes. The ground surveillance was completed from 13–30 August 2019 during the rainy season when the water supply on El Vergel village was not continuous. The village has a community water well with local agreements for supplies. In general, cisterns, cement washbasins (large), and elevated storage tanks were the containers that had more water storage for domestic uses. The water on roofing puddles could not be determined by ground surveillance.

### 3.2. Drone Surveillance (DS) Method

DS found a total of 983 containers in the 216 households in El Vergel, Tapachula, during the same period of GS. In addition, only 10 of the 26 types of containers (38.4%) were detected by GS: disposable plastic containers, plastic buckets and tubs, elevated storage tanks, cement washbasins, tires, toilets, flowerpots, tin buckets, glass bottles and others (Table 2). Of the 983 containers, 147 (15%) were observed in roofs, while 836 (85%) were in backyards. In general, the most detected containers by DS were plastic buckets and tubs (40.5%), disposable plastic containers (27.8%) and flowerpots (15.8%) (Table 2). Additionally, DS seen with a major frequency were disposable plastic containers on roofs (37.4%) and other types of containers (30.6%), while in backyards, they were plastic buckets and tubs (45.2%), disposable plastic containers (26.1%) and flowerpots (18.2%) (Table 2). Figure 2 and Figure 3 show that the resolution of the 20 MP used in the drone camera was enough to distinguish and account for the different container types present in house environments, specifically on roofs and in backyards. However, the canopy of vegetation or shadows generated by walls can hide containers. Drone surveillance was completed from 13 August–6 September 2019, and had a sequential order when taking pictures.

### 3.3. Comparison between Drone and Ground Surveillance

Due to the resolution of 20 MP not being enough to distinguish whether a container held water in the pictures, the comparisons between DS and GS considered all containers regardless of whether they carried water or not and we used proportions instead of percentages for this propose. The overall results indicate that the proportion of all containers detected by DS and GS was 1:2.8, with approximately one container detected by DS for each three containers detected by GS. The container types with a higher proportion DS includes DG, which means that DS detected more containers than GS, and included disposable plastic containers (1.25:1), plastic buckets and tubs (2.08:1), tin buckets (2.20:1) and others (1.49:1). The lower proportion DS includes DG and were cement washbasins (1:13.5), toilets (1:7.3), flowerpots (1:5.5) and glass bottles (1:4.3) (Table 2). The only type of container that the drone was able to find on roofs to almost the same extent as GS was “other”, with a proportion of 1.05:1. However, disposable plastic containers (1:1), plastic buckets and tubs (1.98:1) and tin buckets (2:1) were more frequently observed in backyards by DS than GS. A total of 197 backyards containers (10 tin buckets and 187 plastic buckets and tubs) and 135 roof containers (except 12 elevated storage tanks), were detected by DS, but were not detected by GS (Table 2). It is necessary to mention that 78% of the study dwellings had metal roofing sheets, making it hard to identify breeding sites by GS.

The low camera resolution previously mentioned also meant that this study could not corroborate the presence of every container in both DS and GS; however, we could establish the percentage of the presence/absence of each container type in households, which are expressed in Table 3. The drone was able to observe at least one container in 146 of the 216 households (67.6%), while GS observed 197 (91.2%). Moreover, both methods together could detect at least one container in 208 households (96.3%); however, only in 135 (64.9%) were they concordant in both DS and GS. These results mean that DS only increased its ability by 5.1% over traditional vector surveillance to detect houses with at least one container. Nonetheless, a deeper analysis of container types detected by DS showed that plastic and bucket tubs (46.8%) and disposable plastic containers (28.7%) were the only type that the drone could observe more frequently than GS in household environments (Table 3). Meanwhile, the drone was not as efficient as GS for the rest of the container types in observing the lower presence percentages (Table 3). Additionally, Kendall’s analysis, which is also used for presence/absence data, revealed weak correlation values for every container type: plastic containers (*p* = 0.3464), plastic buckets and tubs (*p =* 0.3885), storage tanks (*p =* 0.4791), cement washbasins (*p =* 0.2700), tires (*p =* 0.0758), toilets (*p =* 0.6260), flowerpots (*p =* 0.3557), tin buckets (*p =* 0.1304), glass bottles (*p =* 0.6817), and others (*p =* 0.7911).

## 4. Discussion

Container type, location, and stored water depends on many socioeconomic factors, including city water supply services, in addition to region or country customs. Inhabitants are accustomed to storing water in unlidded washbasins and other containers, and so support immature mosquito development. Concurrently, surrounding tree shadows and house characteristics allow for the perfect scenario for the mosquito’s life-cycle. Clearly, early assessment of related ecological factors has been the surveillance priority of all government vector control programs. However, traditional field sampling operations for this task mostly rely on ground or house visits.

This is a very time-consuming and an expensive duty with doubtful and temporary results. Early attempts were made to evaluate satellite photography as a surveillance tool in identifying residential premises at a high risk of *Aedes aegypti* infestations [26]. Studies concluded that the ability of low-level aerial photography to enhance *Aedes aegypti* breeding sites for surveillance was limited and ground surveillance was the most reliable tool to identify the probability of *Aedes aegypti* breeding in the residential environment [26]. However, the latter study opened the way to test new technological tools to obtain higher resolution photos that generated stronger entomological results. In addition, drones can be a surveillance tool to identify breeding sites in inaccessible areas of the home, such as roofs, and in closed houses and vacant lots; this is due to the importance of monitoring these areas of mosquito populations [31]. This is particularly the case in areas with COVID-19 cases, where health personnel should not be in contact with inhabitants [15,32]. Currently, evidence of drone effectiveness for identifying mosquito breeding sites is scarce, and it is based on *Anopheles* mosquitos in rural or semirural areas, and in areas with more extensive and different landscapes [33,34].

During this study, by using drone images, we were able to identify the presence of containers in 146 of the 216 households surveyed (67.6%), while the traditional grounds methods reached 197 (91.2%). Moreover, the concordance between both DS and GS was 64.9%, showing the presence of a container in 135 households, but the correlations of both methods for each container type were weak. In addition, the number of containers were clearly higher with GS, as we detected 1 container by DS per 2.8 by GS. Additionally, we identified 10 out of the 26 container types by DS, where the most frequent containers in backyards were plastic buckets and tubs (378, 45.2%), disposable plastic containers (218, 26.1%) and flowerpots (152, 15.4%), while the roof containers that were observed on more occasions by drones were disposable plastic containers (55, 37.4%) and others (45, 4.6%). The lower number of containers detected by drones may be explained by the presence of ceilings and trees, in addition to house walls and plastic sheets that do not allow for the observation of containers. We had the same visibility limitations as were present in similar studies in Australia and New York [25,26]; to improve the aerial surveillance of yard condition components, one could seek either a larger scale image and/or an image taken closer to midday, which would reduce shadowing [26]. In our study, a shadow was minimal with our drone flights being at noon, where the day temperature limited the length of the flights; this was according to the manufacture’s advice that the operating temperature range was 32 °F to 104 °F (0 °C to 40 °C) [23]. Moreover, the low resolution of the camera used in this investigation may also explain why we could not find more containers using DS. According to a previous study [25], the camera also did not take reliable images of small containers. We propose to explore other imaging techniques outside of the visible spectrum, such as multispectral technologies, in conjunction with the effectiveness of UAVs, in order to image potential mosquito habitats to identify water retention areas that can be used to identify probable mosquito breeding places [26,27].

In a recent study on *Aedes albopictus*, a convolutional neural network using drone images could identify potential breeding habitats with a precision of 67% [25]. We recommend that further studies extend the application of neural networks for *Aedes aegypti* surveillance [25,35]. However, neural network technology requires numerous images of many different scenarios and a significant investment of time in order to increase precision. More cost-effective studies are needed to identify the utility of drones in different scenarios and countries.

## 5. Conclusions

The present study is the first in Mexico to study the use of drones to identify *Aedes aegypti* mosquito breeding sites. By DS, we identified a total of 983 containers distributed in 10 types, which is approximately one-third of the 2752 containers that were inspected by ground surveillances and distributed in 26 types. The concordance between drone and the ground surveillance was 64.9% in detecting the presence of any container by both methods; however, the correlation values for each container type were weak. The most valuable finding of this research was that drones could identify potential breeding sites, such as plastic buckets and tubs, disposable plastic containers, flowerpots, and others, in inaccessible household areas that traditional ground surveillance could not reach. Drones can be useful in mosquito surveillance and control programs as a complementary tool and can be used in hard-to-access areas, vacant houses/lots, schools, and public buildings in order to involve families, communities, authorities, and different actors to enhance healthy environments. Drone technology can be applied especially in homes with confirmed cases of COVID-19, where health workers should not perform activities inside the house. Additionally, drones can be used after natural disasters in regards to the appearance of dengue outbreaks in order to strengthen entomological surveillance in disaster areas and/or during a health emergency, and as part of the decision-making team in public health in the Emergency and Disaster Committees. Drone utility deserves to be further explored and cost-effectiveness studies need to be made on products like specialized cartography, surveillance, mosquito control, and the release of sterile mosquitoes as an acceptance of community studies and a legal framework to apply these technologies.

## Figures and Tables

**Figure 1 insects-12-00663-f001:**
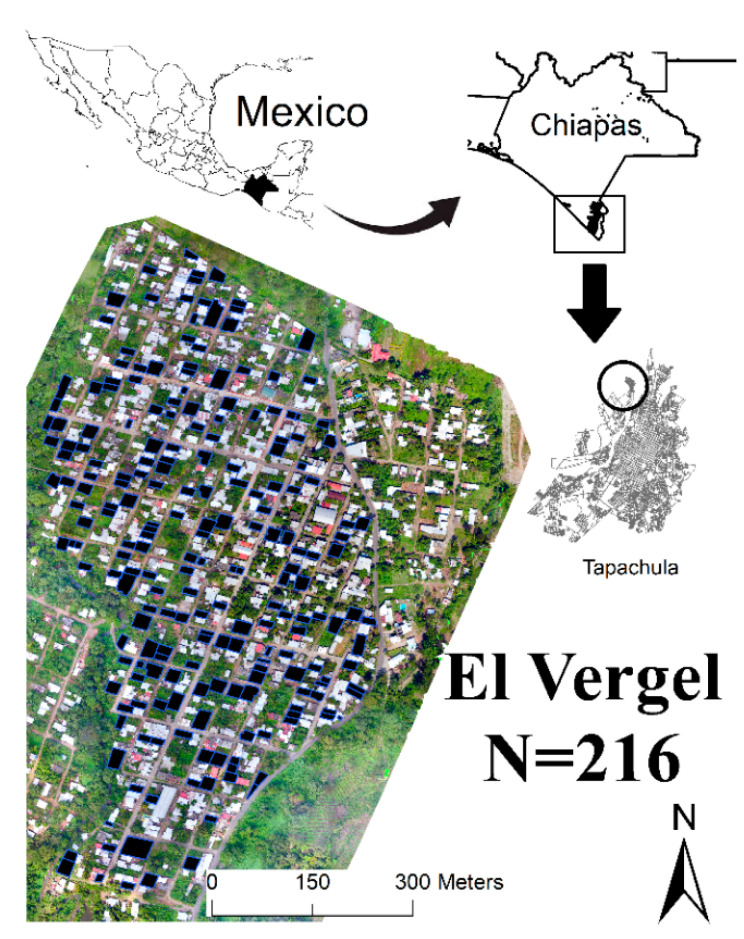
Geographical representation of the study area, El Vergel neighborhood, north of Tapachula city and municipality, in the far southeast of Chiapas, Mexico. The black rectangles represent the study dwellings.

**Figure 2 insects-12-00663-f002:**
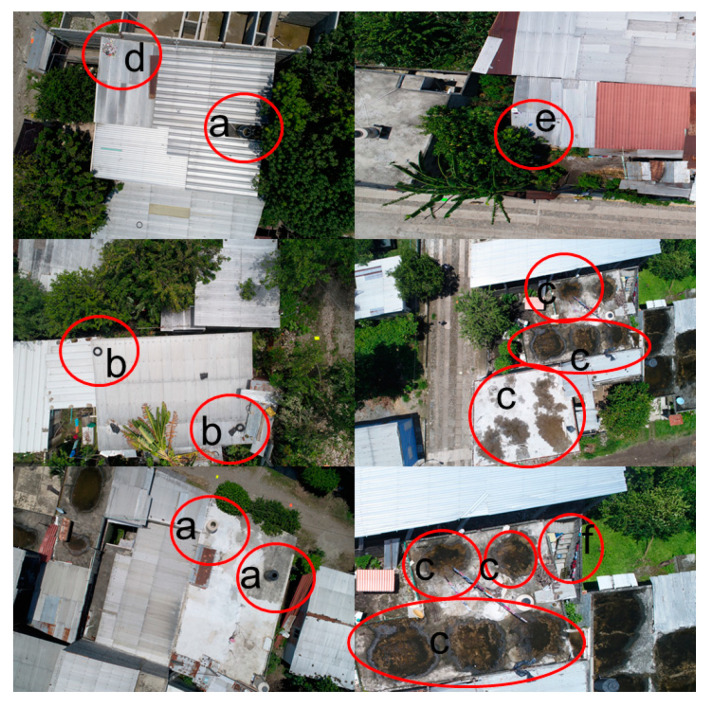
Drone images of larvae breeding sites on the roofs of houses in El Vergel, Tapachula, Chiapas, in (**a**) elevated storage tank; (**b**) tires; (**c**) others (puddles); (**d**) disposable plastic containers, (**e**) flowerpots and (**f**) plastic bucket & tubs.

**Figure 3 insects-12-00663-f003:**
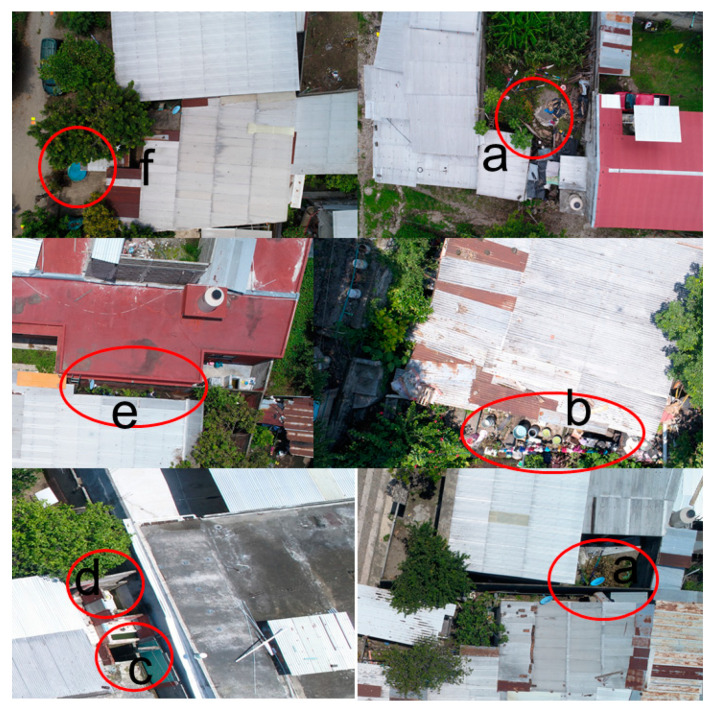
Drone images of larvae breeding sites in outdoor (backyard) areas of houses in El Vergel, Tapachula, Chiapas, in (**a**) disposable plastic container; (**b**) plastic buckets and tubs; (**c**) cement washbasins (large); (**d**) WCs; (**e**) flowerpots; and (**f**) other (pools).

**Table 1 insects-12-00663-t001:** *Aedes aegypti* larvae breeding sites found in the El Vergel, Tapachula, Chiapas, 13–30 August 2019, using the traditional field technician’s surveillance (ground surveillance). Containers by house location and positive containers for larvae presence for every 26 categories.

Container Type	House Location				Total	
	Indoor (%)	+/−Larvae Presence (%)	Outdoor (%)	+/−Larvae Presence (%)	Frequency (%)	+/−Larvae Presence (%)
Plastic bucket & tubs	60 (14.4)	3 (5.0)	191 (8.2)	12 (6.3)	251 (9.1)	15 (6.0)
Tin bucket	2 (0.5)	1 (50.0)	10 (0.4)	1 (10.0)	12 (0.4)	2 (16.7)
55-Ga drum	31 (7.4)	5 (16.1)	55 (2.4)	12 (21.8)	86 (3.1)	17 (19.8)
Non-disposable plastic container	51 (12.2)	2 (3.9)	232 (9.9)	10 (4.3)	283 (10.3)	12 (4.2)
Non-disposable glass container	17 (4.1)	0 (0.0)	51 (2.2)	1 (2.0)	68 (2.5)	1 (1.5)
Non-disposable pewter container	1 (0.2)	0 (0.0)	3 (0.1)	1 (33.3)	4 (0.1)	1 (25.0)
Clay pot	1 (0.2)	0 (0.0)	2 (0.1)	1 (50.0)	3 (0.1)	1 (33.3)
Cement washbasin (large)	97 (23.2)	42 (43.3)	108 (4.6)	37 (34.3)	205 (7.4)	79 (38.5)
Cement washbasin (small)	2 (0.5)	0 (0.0)	6 (0.3)	1 (16.7)	8 (0.3)	1 (12.5)
Elevated storage tank	2 (0.5)	0 (0.0)	36 (1.5)	3 (8.3)	38 (1.4)	3 (7.9)
Cistern	0 (0.0)	0 (0.0)	2 (0.1)	0 (0.0)	2 (0.1)	0 (0.0)
Flowerpot	23 (5.5)	0 (0.0)	846 (36.2)	6 (0.7)	869 (31.6)	6 (0.7)
Animal drinking container (fixed)	5 (1.2)	0 (0.0)	24 (1.0)	1 (4.2)	29 (1.1)	1 (3.4)
Animal eating container (fixed)	0 (0.0)	0 (0.0)	4 (0.2)	0 (0.0)	4 (0.1)	0 (0.0)
Glass bottle	1 (0.2)	0 (0.0)	39 (1.7)	1 (2.6)	40 (1.5)	1 (2.5)
Can	0 (0.0)	0 (0.0)	151 (6.5)	3 (2.0)	151 (5.5)	3 (2.0)
Animal drinking (mobile)	1 (0.2)	0 (0.0)	141 (6.0)	5 (3.5)	142 (5.2)	5 (3.5)
Animal eating container (mobile)	0 (0.0)	0 (0.0)	2 (0.1)	0 (0.0)	2 (0.1)	0 (0.0)
Disposable plastic container	7 (1.7)	0 (0.0)	218 (9.3)	4 (1.8)	225 (8.2)	4 (1.8)
Disposable glass container	2 (0.5)	0 (0.0)	11 (0.5)	0 (0.0)	13 (0.5)	0 (0.0)
Disposable pewter container	0 (0.0)	0 (0.0)	3 (0.1)	1 (33.3)	3 (0.1)	1 (33.3)
Vase	14 (3.3)	3 (21.4)	4 (0.2)	1 (25.0)	18 (0.7)	4 (22.2)
Tire	11 (2.6)	0 (0.0)	84 (3.6)	7 (8.3)	95 (3.5)	7 (7.4)
Toilet	82 (19.6)	1 (1.2)	66 (2.8)	2 (3.0)	148 (5.4)	3 (2.0)
Discarded stove	0 (0.0)	0 (0.0)	2 (0.1)	0 (0.0)	2 (0.1)	0 (0.0)
Others	8 (1.9)	2 (25.0)	43 (1.8)	8 (18.6)	51 (1.9)	10 (19.6)
Total (*n*)	418 (100.0)	59 (14.1)	2334 (100.0)	118 (5.1)	2752 (100.0)	177 (6.4)

**Table 2 insects-12-00663-t002:** Total number of containers detected by drone (DS) and ground surveillance (GS) as potential breeding sites for *Aedes aegypti* in the locality of El Vergel, Tapachula, Mexico, 13 August–6 September 2019. Only 10 of the 26 container types were detected by drone surveillance; therefore, data of ground surveillance from these types of containers was considered in this table for comparison proposes.

Container Type	Drone Surveillance (DS)			Ground Surveillance (GS)	Proportion of Containers DS in Roofs:GS	Proportion of Containers DS in Backyards:GS	Proportion of Containers DS Total: GS
	Roofs (%)	Backyards (%)	Total (%)				
Disposable plastic containers	55 (37.4)	218 (26.1)	273 (27.8)	218	1:4.0	1:1	1.25:1
Plastic bucket & tubs	20 (13.6)	378 (45.2)	398 (40.5)	191	1:10.0	1.98:1	2.08:1
Elevated storage tanks	12 (8.2)	8 (1.0)	20 (2.0)	36	1:3	1:4.5	1:1.8
Cement washbasins (large)	0 (0.0)	8 (1.0)	8 (0.8)	108	0:1	1:13.5	1:13.5
Tires	10 (6.8)	15 (1.8)	25 (2.5)	84	1:8.4	1:5.6	1:3.4
Toilets	0 (0.0)	9 (1.1)	9 (0.9)	66	0:1	1:7.3	1:7.3
Flowerpots	3 (2.0)	152 (18.2)	155 (15.8)	846	1:282	1:5.6	1:5.5
Tin buckets	2 (1.4)	20 (2.4)	22 (2.2)	10	1:5	2:1	2.20:1
Glass bottles	0 (0.0)	9 (1.1)	9 (0.9)	39	0:1	1:4.3	1:4.3
Others	45 (30.6)	19 (2.3)	64 (6.5)	43	1.1:1	1:2.3	1.49:1
Total (*n*)	147 (15.0)	836 (85.0)	983 (100)	2752 *	1:18.7 *	1:3.3 *	1:2.8 *

* Considering all containers detected by ground surveillance.

**Table 3 insects-12-00663-t003:** Percentages of households with the presence or absence of potential breeding sites of *Aedes aegypti,* as determined by drone (DS) and ground (GS) surveillance in 216 households from the locality El Vergel, Tapachula, Mexico, August 2019. Only 10 out of the 26 container types were detected by drone surveillance; therefore, data of ground surveillance from these types of containers was considered in this table for comparison purposes.

Container type	Presence by DS (%)	Absence by DS (%)	Presence by GS (%)	Absence by GS (%)	Presence DS + GS (%)	Absence DS + GS (%)	Concordance DS–GS (%) *
Disposable plastic containers	62 (28.7)	154 (71.3)	18 (8.3)	198 (91.7)	77 (35.6)	139 (64.3)	3 (3.9)
Plastic bucket & tubs	101 (46.8)	115 (53.2)	84 (38.9)	132 (61.1)	142 (65.7)	74 (34.3)	43 (30.3)
Elevated storage tanks	8 (3.7)	208 (96.3)	34 (15.7)	182 (84.3)	40 (18.5)	176 (81.5)	2 (5.0)
Cement washbasins (large)	7 (3.2)	209 (96.8)	106 (49.1)	110 (50.9)	108 (50.0)	108 (50.0)	5 (4.6)
Tires	9 (4.2)	207 (95.8)	16 (7.4)	200 (92.6)	23 (10.6)	193 (89.4)	2 (8.7)
Toilets	8 (3.7)	208 (96.3)	64 (29.6)	152 (70.4)	69 (31.9)	147 (68.1)	3 (4.35)
Flowerpots	48 (22.2)	168 (77.8)	101 (46.8)	115 (53.2)	127 (58.8)	89 (41.2)	22 (17.32)
Tin buckets	16 (7.4)	200 (92.6)	10 (4.6)	206 (95.4)	24 (11.1)	192 (88.9)	2 (8.33)
Glass bottles	3 (1.4)	213 (98.6)	12 (5.6)	204 (94.4)	15 (6.9)	201 (93.1)	0 (0.0)
Others	19 (8.8)	197 (91.2)	39 (18.1)	177 (81.9)	55 (25.5)	161 (84.5)	3 (5.5)
Total (*n*)	146 (67.6)	70 (32.4)	197 (91.2)	19 (8.8)	208 (96.3)	8 (3.7)	135 (64.9)

* Number and percentages of houses with at least one container detected by both DS and GS.

## Data Availability

Not applicable.

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
