# Peer review of "Field Effectiveness of Drones to Identify Potential Aedes aegypti Breeding Sites in Household Environments from Tapachula, a Dengue-Endemic City in Southern Mexico"

_insects, 2021, doi:10.3390/insects12080663_

Round 1
Reviewer 1 Report
This is a very interesting paper and it demonstrates two opposite conclusions simultaneously: 1, nothing can replace "boots on the ground" for locating mosquito larval habitats; 2, drones are valuable surveillance tools and should be incorporated into inspection programs. The use of drones will alleviate the need to climb onto roofs of houses and should increase speed at which inspectors can search a neighborhood.
I do not like the use of the term "breeding" in place of "immature" but that is a minor point.
Something that the authors might consider: they might add another statistical analysis to their paper. Using Kendall's tau statistic (basically a rank-order correlation) might reveal whether certain kinds of containers were located more often via ground-based inspection or via drones. This might give some insight as to how to allocate effort between a drone team and a ground team.
There are some instances where the English usage is stilted or not exactly correct. The authors should have a native speaker read through the manuscript and correct some of the construction. For example, in line 193 the authors write of containers "in roofs" whereas in English it is much more common to write that something is "on" a roof; "in" a roof implies to an English-speaking reader that something is embedded into the roofing material (and this might actually be the case if the roof is covered in tar). There are a number of these awkward constructions and they detract from the paper, which is a very good paper. Overall this paper is well-written but it needs some clean up - right now it looks as if the authors are "painting with a dirty brush".
Author Response
Thanks for carefully reviewing our manuscript. The point-point response was as attached

Reviewer 2 Report
Dear Authors
This study is very interesting and provides new insights on ways to continue vector surveillance activities even in times of Covid 19 pandemics. In addition, it shows that drone utility deserves to be further explored and that with some technological and operating improvements, it can be a valuable tool to fight mosquito-borne diseases. However, a careful revision of the manuscript is needed in order to improve the grammar quality.
Some suggestions for revision:
Simple Summary, line 20 where we read “effectiveness”, we should read “effective”;
Simple Summary, line 22 where we read “is the first report that use drones”, maybe we should read “is the first reporting the use of drones”;
Simple Summary, line 25 where we read “in the roof and backyard households”, we should read “in the households’ roofs and backyards”;
Simple Summary, line 49 where we read “and therefore, a better”, we should read “and therefore, better”;
Introduction, line 57 the authors use “DF” for the first time and its definition (Dengue fever) is in only on line 59;
Introduction, line 61 where we read “diurnal rhythms of feeding observing the highest biting activity in mid-morning” we should read “diurnal rhythms of feeding and the highest biting activity is observed in mid-morning”;
Introduction, line 75 where we read “y/o” we should read “and/or”;
Introduction, line 89 where we read “further analized” we should read “further analizing them”;
Introduction, line 95 where we read “in the roof and backyard households”, we should read “in the households’ roofs and backyards”;
Figure 1 should be improved. We can hardly see the “a)” in the picture on the right;
Materials and Methods, lines 131 and 132 where we read “because can difficult”, we should read “because this can hamper the”;
Materials and Methods, line 152 it is the first time the authors mention DS and GS. The abbreviations should be explained here;
Results, line 234 please italicize and correct “Aedes aeygpti”;
Results, line 238 where we read “These lasts”, we should read “These results”;
Results, line 240 where we read “by DS gave”, we should read “by DS showed”;
Discussion, line 269 where we read “During study”, we should read “During this study”;
Discussion, line 285 where we read “visibility is clearly”, we should read “visibility being clearly”;
Discussion, line 300 where we read “we recommended for future studies, extend”, we should read “we recommend for further studies to extend”;
Discussion, lines 302 to 306, please rephrase because there are several grammatical incorrections;
Discussion, last paragraph, please rephrase because there are several grammatical incorrections.
Author Response
Thanks for carefully reviewing our manuscript. The point-point response was as attached:

Reviewer 3 Report
While I express appreciation for the tentative drone application you have done, the results are quite unsatisfactory and do not support your positive attitude in the conclusions. It seeems that the drone technology must be strongly improved to achieve some kind of usefulnes in Aedes aegypti surveillance and control. Table 3 is confusing and should be eliminated or re-organized.
While I express my appreciation for the tentative drone application you have done, the results are quite unsatisfactory and do not support the positive attitude you included in the conclusions. It seems that the drone technology must be strongly improved to be able to achieve some kind of usefulness in Aedes aegypti surveillance and control. In its current capacity, I do not see convenience in its adoption. Table 3 is confusing and should be eliminated or re-organized. Some comments are included into the manuscript.

Author Response

(The authors gave the same response as above.)
